# Directed exciton transport highways in organic semiconductors

Kai Müller[1,2], Karl S. Schellhammer[1,3], Nico Gräßler [3,4], Bipasha Debnath[4], Fupin Liu [4], Yulia Krupskaya[4], Karl Leo[3], Martin Knupfer[4] & Frank Ortmann [1,5] ✉

Exciton bandwidths and exciton transport are difficult to control by material design. We showcase the intriguing excitonic properties in an organic semiconductor material with specifically tailored functional groups, in which extremely broad exciton bands in the near-infrared-visible part of the electromagnetic spectrum are observed by electron energy loss spectroscopy and theoretically explained by a close contact between tightly packing molecules and by their strong interactions. This is induced by the donor–acceptor type molecular structure and its resulting crystal packing, which induces a remarkable anisotropy that should lead to a strongly directed transport of excitons. The observations and detailed understanding of the results yield blueprints for the design of molecular structures in which similar molecular features might be used to further explore the tunability of excitonic bands and pave a way for organic materials with strongly enhanced transport and built-in control of the propagation direction.

Organic semiconductors offer a multitude of novel electronic device concepts with attractive parameters such as flexibility, low weight and cost, and biocompatibility[1]. However, for many applications, the comparatively poor charge carrier and energy transport in these carbon-based materials remains a road block. For instance, energy capture and efficient energy transfer are key processes for solar energy conversion in natural and artificial molecular systems[2,3]. In organic photovoltaic films, the solar energy is carried to the internal interfaces by diffusive excitons, which generate charges and eventually deliver electrical power. The poor diffusivity of the excitons in organic films is critical because slow migration prevents the arrival at an interface within the exciton lifetime[4]. Since a significant improvement of exciton diffusion could not be achieved in the past, disordered bulk heterojunction (BHJ) morphologies were introduced[5,6] to increase the photo-currents and efficiencies of organic solar cells and organic photodetectors[7–9]. However, BHJ film morphologies, are difficult to understand[10] and control, and short exciton diffusion lengths further restrict the optimization of BHJ devices (e.g. by domain sizes)[4,11]. Strongly improved exciton diffusion lengths or a greater exciton delocalization at room temperature are sought-after[12,13] because this should allow novel design principles or add new directions for optimization of photovoltaic blends. This intriguing perspective opens the quest for broad exciton bands allowing faster and better controlled exciton transport[14]. However, while exciton bandwidths were initially predicted to reach 0.2 eV[15], this value is exceeded by only a few experimental systems[16] or for systems with a very large optical gap (vide infra). The exciton bandwidth often remains clearly below the electronic bandwidth of benchmark organic crystals[17]. Despite encouraging progress for polythiophene-based nanofibres[11,18], it remains difficult to observe delocalized excitons at room temperature[13] and to control their pathways[18,19].

In this work, we show through theoretical and experimental studies of the quinoid merocyanine QM1 (see Fig. 1a) that it breaks with this paradigm. We study the intriguing excitonic features in thin films

[1]Center for Advancing Electronics Dresden, Technische Universität Dresden, 01062 Dresden, Germany. [2]Institut für Theoretische Physik, Technische Universität Dresden, 01062 Dresden, Germany. [3]Dresden Integrated Center for Applied Physics and Photonic Materials (IAPP) and Institute for Applied Physics, Technische Universität Dresden, 01062 Dresden, Germany. [4]Leibniz Institute for Solid State and Materials Research Dresden, Helmholtzstr. 20, 01069 Dresden, Germany. [5]Department of Chemistry, TUM School of Natural Sciences, Technische Universität München, Lichtenbergstr. 4, 85748 Garching b. München, Germany. ✉e-mail: frank.ortmann@tum.de

by performing momentum-resolved electron-energy-loss spectroscopy (EELS). We have further grown QM1 single crystals, determined the crystal structure, and performed comprehensive theoretical investigations using exciton models and density functional theory (DFT) calculations to simulate the exciton band structure and dielectric properties. We find an extremely large exciton bandwidth for the low-energy excitations of about 1.33 eV, which exceeds typical values in organic semiconductors by far. This bandwidth results from the intermolecular coupling that forms quasi-one-dimensional unidirectional exciton pathways. Moreover, transition–quadrupole and transition–octupole exciton channels with smaller dispersion have been discovered. The experimental results are consistently described by the theoretical models, which leads to fundamentally new insights, demonstrating the richness of excitonic coupling, and paving the way for exciton engineering in organic solids.

## Results

### Material characterization and electron energy loss spectroscopy

Quinoid merocyanines (QM) have been utilized in vacuum-processed organic solar cells due to their intensive absorption and high thermal stability[20, 21]. The dye QM1 ((E)-2-(5′-(1,3-dithiol-2-ylidene)-5H,5′H-[2,2′-bithiophenylidene]-5-ylidene)-malononitrile) shown in Fig. 1a has a broad thin-film absorption band extending up to 1100 nm (1.1 eV, Fig. 1b) that is promising for strong excitonic coupling and potentially yields wide exciton bands. Exciton energies can be directly probed by EELS as a function of transferred quasi-momentum $q \equiv |\mathbf{q}|$ and, in this way, information on the dispersion (energy-momentum behavior) or the size of the wave function, representing an excitation, can be studied[22,23]. Organic thin films of QM1 were prepared as explained in the "Methods" section. We note that with our high primary beam energy only singlet excitations are probed. The energy and

momentum resolution was chosen to be 85 meV and 0.04 Å$^{-1}$, respectively. Momentum values have been chosen such that they do not correspond to Bragg peak positions in the diffraction profile in order to minimize multiple scattering effects. All data have been measured at low temperatures (20 K) in order to enhance sample lifetime as well as to avoid thermal broadening of the linewidths[24].

Figure 1c shows the EELS spectra, which exhibit strong changes upon variation of the exciton wave number $q$ in the whole energy range observed. The main resonance (feature $F_1$) at energy $E_1$ shifts strongly to the red with increasing $q$ and we quantify this shift to 1 eV between the peaks for $q = 0.1$ and $q = 0.75$ Å$^{-1}$. In addition, at small ($q = 0.1$ Å$^{-1}$) and large ($q = 0.75$ Å$^{-1}$) momenta, shoulder features $F_1'$ and $F_2$ are visible. Even without the $F_2$ shoulder, the full-width-at-half-max of this low-energy exciton band is determined to be 1.33 eV. Intriguingly, another strong resonance $F_3$ is observed at $E_3$ ($q = 0.75$ Å$^{-1}$) = 3.1 eV and exhibits quite peculiar behavior. In contrast to $F_1$, $F_3$ shows only weak energy dispersion. In addition, it emerges from the background with increasing $q$ and is absent in the optical limit ($q \to 0$). In consistency at zero wave vector, the UV–Vis absorption does not show any related feature (Fig. 1b) at about 3 eV. We finally observe a fourth feature $F_4$, which emerges also with increasing $q$ and hardly changes position with increasing $q$ (similar to $F_3$), suggesting a relationship between $F_3$ and $F_4$, but not to $F_1$.

To connect these observations in thin film with the molecular structure, the electronic and optical properties of QM1, molecules are simulated with DFT (see Supplementary Methods for simulation details). First, we observe a very strong permanent dipole moment of 17.8 D for QM1, which can be rationalized by the strong push-pull character of the molecule imposed by dicyanomethylene accepting (A) and dithiolene donating (D) groups. We also find a large oscillator strength of $v = 1.33$ for the lowest-energy optical transition in time-dependent DFT (TDDFT) and a small molecular relaxation energy for this exciton of $\Lambda = 130$ meV. The transition dipole (TD) moment points along the long axis of the molecule and we find that the principle dipole transitions are due to transitions between the highest occupied molecular orbital (HOMO) and the lowest unoccupied molecular orbital (LUMO) (see Supplementary Table 2 for a characterization of other optical transitions).

### Crystal structure and molecular packing

In order to investigate the molecular interactions, we grow single crystals of QM1 (see Methods section, crystal data in Supplementary Table 1). Fig. 2b, d shows the tight packing of QM1 molecules with 4 molecules per simple unit cell (space group $P\bar{1}$), which is markedly different from the herring-bone stacked acenes or thiophenes. Laterally, the molecules form dimers with antiparallel alignment (e.g. molecules (1,2) in Fig. 2b) to cancel the net permanent molecular dipole moment.

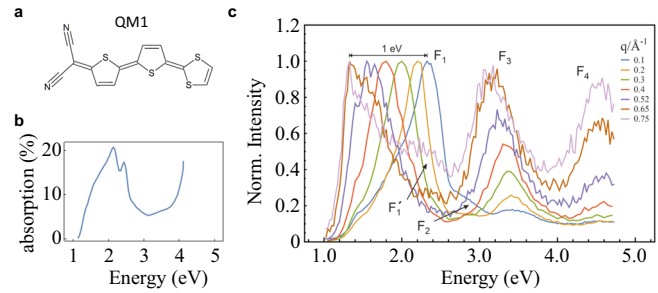

**Fig. 1 | Material and excitation spectra. a** Chemical structure of QM1 used in the study. **b** Optical absorption of a 20 nm film of QM1[20]. **c** Normalized experimental EELS spectra. The peak height of $F_1$ is used for normalization.

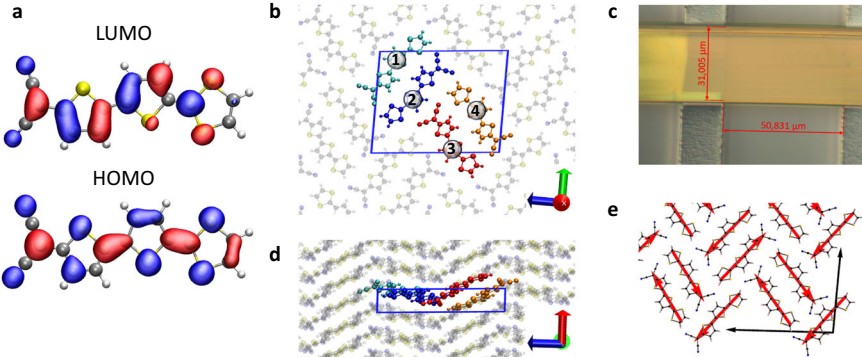

**Fig. 2 | Structure, crystal packing and transition dipoles. a** HOMO and LUMO structures obtained from DFT calculations (M06-2X/cc-pVTZ). **b, d** Top view and side view of the QM1 crystal structure. Unit cell is indicated in blue. **c** Photograph of a QM1 crystal. **e** Arrangement of transition dipole moments (red arrows) relative to the molecules in the crystal structure.

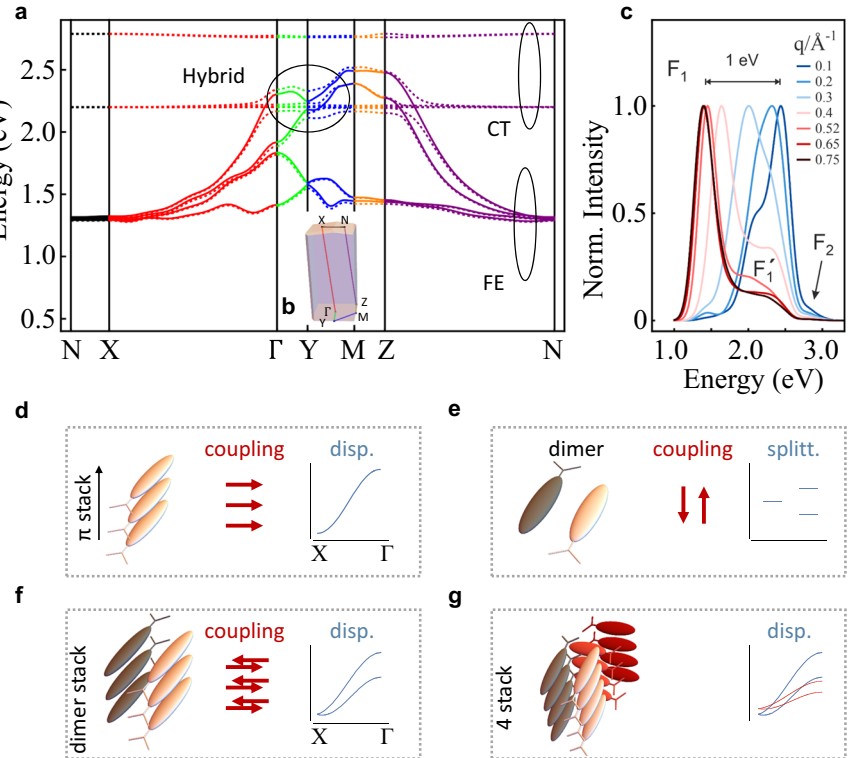

**Fig. 3 | Low-energy exciton features. a** Low-energy exciton band structure $E_\nu(\mathbf{q})$ along the directions between special points in the Brillouin zone. The exciton band structure of the pure Frenkel model is plotted as a solid line and the exciton band structure including Frenkel (FE) and charge transfer (CT) excitons is plotted in dashed lines. **b** Brillouin zone with the definition of special points. **c** Simulated EELS spectrum for the QM1 crystal of Fig. 2. **d–g** Relation between molecular arrangement (molecular π stack, dimer, dimeric stack or 4-mer stack), their TD coupling (red arrows; parallel or anti-parallel) and the resulting features in the energy dispersion for the ΓX direction, which rationalizes the band structure qualitatively.

This arrangement further takes advantage of directional hydrogen bonds between the molecules, similar to DNA base crystals[25]. More importantly, in the vertical direction, the dimers are π-stacked in a parallel-displaced ladder-like packing which allows for a short π–π distance. Interestingly, neighboring dimer pairs (with molecules (3,4) in Fig. 2b, d) are rotated by about 77° and stacked differently.

**Exciton modeling**

To understand the exciton characteristics in crystalline QM1, we use an exciton model based on local molecular excitons (ME) and charge-transfer (CT) states[26], capturing the essential interactions. In organic materials, the HOMO−LUMO transitions are often the most important ones at low excitation energy, which is also the case for QM1 here (see Supplementary Table 2). We can therefore initially restrict the model to the HOMO and LUMO orbitals that are located on each molecule. An extension beyond this set of orbitals will be discussed further below for higher energies. The electron−hole pair states (or exciton states) are hence described on the basis of electron and hole positions $|\mathbf{r}_{hole}, \mathbf{r}_{electron}\rangle$ according to

$$|\Phi_\Lambda\rangle = \sum_{\mathbf{n}, j, \mathbf{n}', j'} \Phi_\Lambda(\mathbf{n}, \mathbf{r}_j, \mathbf{n}', \mathbf{r}_{j'}) |\mathbf{n} + \mathbf{r}_j, \mathbf{n}' + \mathbf{r}_{j'}\rangle, \quad (1)$$

where the coefficients $\Phi_\Lambda(\mathbf{n}, \mathbf{r}_j, \mathbf{n}', \mathbf{r}_{j'})$ are known as exciton amplitude ($\mathbf{n}$ and $\mathbf{n}'$ are unit cell indexes and $\mathbf{r}_j$ the molecular positions). The simplest family of models, which is frequently and successfully used in literature, considers only local MEs[27], i.e. $\Phi_\Lambda(\mathbf{n}, \mathbf{r}_j, \mathbf{n}', \mathbf{r}_{j'}) \rightarrow \delta_{\mathbf{n}\mathbf{n}'} \delta_{jj'} \Phi_\Lambda(\mathbf{n}, \mathbf{r}_j, \mathbf{n}, \mathbf{r}_j)$, which leads to the Frenkel model. However, it has been shown in organic systems that molecular excitons may hybridize with CT excitons[28–31]. Therefore, we go beyond the Frenkel model[32,33] by including excitons of CT character and account for the coupling

between these different manifolds (see the "Methods" section and Supplementary Methods).

Owing to the periodicity of the system, we solve for the excitation energies $E_\nu(q)$ (with band index $\nu$) and obtain the exciton band structure (Fig. 3). We find an extremely strong energy dispersion $E_\nu(q)$ which results in a very large exciton bandwidth. When measured between the minimum along XΓ and the maximum at point M in the Brillouin zone, the theoretical bandwidth reaches 1.25 eV. This is much larger than the molecular relaxation energy (130 meV) for this exciton state and implies a coherent exciton delocalization over many molecules at room temperature. Note that delocalization was discussed and predicted even for systems with smaller bandwidths[12,13] and our findings are consistent with recent insights into electron−phonon coupled systems[34] and exciton−phonon coupling[35] as well as findings on other systems[11]. The theoretical bandwidth is also close to the EELS bandwidth of 1.33 eV. We emphasize that this value is almost an order of magnitude larger than typical values[17,23,26,29,30,36–38] and exceeds the largest one observed so far. Fig. 4 summarizes reference values comparing the optical gap energy and exciton bandwidth. Values and references are compiled in Supplementary Table 7. Not only is the bandwidth the largest found so far (similar to sexithiophene, 6 T), but QM1 features also outstanding low exciton energies that are different from the high-energy excitons in the herring-bone (HB) materials. This allows better coverage of the visible part of the spectrum and makes the NIR regime accessible.

These remarkable findings are accompanied by a strong anisotropy in the exciton band structure, indicating strongly directed exciton transport: The excitonic coupling causing this behavior is illustrated in Fig. 3d–g. While the dominating part of the dispersion is due to the π stacking in a single column (Fig. 3d), the splitting between bands is caused by the interaction between the two molecules per dimer (Fig. 3e)

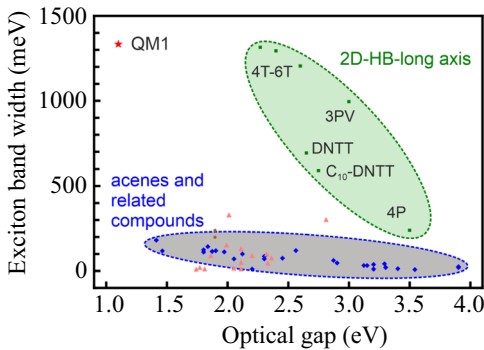

**Fig. 4 | Survey of optical and excitonic properties.** Comparison of 1D excitonic material QM1 with other molecular systems, namely 2D herring-bone (HB) oligomers and acenes and related compounds, which are also arranged in an HB fashion.

in adjacent columns. In addition, the two dimeric stacks in the crystal are coupled leading to a $q$-dependent splitting of the bands (Fig. 3g). The group velocities $v = \frac{1}{\hbar}\frac{dE(q)}{dq}$ are very large, reaching values between 100,000 and 440,000 m/s (among the set of bands) and hence more than 40% of the electron's Dirac point value in graphene. In contrast in the other directions, one observes weaker dispersion or flat bands. The molecular stacks induce an indirect character of the energy gap with the lowest excitation in the near-infrared.

We observe exciton hybridization between molecular and CT excitons in the center of the Brillouin zone. This can be tuned with the wave vector $q$ because the Frenkel-type excitons disperse with $q$, while the charge transfer excitons do not. Therefore much weaker FE–CT hybridization is observed at BZ edge points close to the X or N points. Further experimental and theoretical evidence for FE–CT coupling is discussed below.

## Simulation of EELS spectra

The connection between these features and the EELS experiments is given by the macroscopic dielectric function[39] $\epsilon^M(\omega)$ which describes the system's response to macroscopic electromagnetic fields and is governed by the pair excitations involving occupied and unoccupied states. Their excitation energies are affected by the electron–hole binding that is enhanced in organic materials by the weak screening of the Coulomb interaction between electron–hole pairs and that is properly framed with a Bethe–Salpeter equation[40]. As a result, the macroscopic dielectric function (assuming spin degeneracy and a gapped electronic structure) reads

$$\epsilon^M = 1 + \frac{16\pi}{\Omega}\sum_{\Lambda}^{E_\Lambda > 0}\left|\sum_{\lambda\lambda'}M^*_{\lambda\lambda'}(\mathbf{Q})\Phi_\Lambda(\lambda,\lambda')\right|^2\left[\frac{1}{E_\Lambda - \hbar\omega - i\eta} + \frac{1}{E_\Lambda + \hbar\omega + i\eta}\right],$$
(2)

where $M_{\lambda\lambda'}(\mathbf{Q}) = \frac{e_0}{\sqrt{4\pi\varepsilon_0|\mathbf{Q}|}}\langle\psi_\lambda|e^{i\mathbf{Q}\mathbf{r}}|\psi_{\lambda'}\rangle$ are the transition matrix elements for a complete set of orthogonal single-particle orbitals $|\psi_{\lambda'}\rangle$. $\Phi_\Lambda(\lambda,\lambda')$ is the exciton wave function with excitation energy $E_\Lambda$. $\eta$ is a numerical broadening parameter, and $\Omega$ the volume of the system. With this description, we use an expansion for small wave vectors and introduce the molecular transition dipole moment $\mathbf{d}_j^{HL}$ to first simulate the imaginary part of the dielectric function in transition-dipole approximation

$$\text{Im }\epsilon^M(\omega,\mathbf{q}) = \frac{4}{\Omega}\frac{e_0^2\eta}{\varepsilon_0}\frac{\left|\sum_j e_\mathbf{q}\cdot\mathbf{d}_j^{HL}c_{HL,\mathbf{q}}(\mathbf{r}_j,0)\right|^2}{(E_{HL,\mathbf{q}} - \hbar\omega)^2 + \eta^2},$$
(3)

using the solved crystal structure and the exciton coefficients $c_{HL,\mathbf{q}}(\mathbf{r}_j,0)$ obtained from diagonalization of the pair Hamiltonian in reciprocal space. To compare to the experimental EELS spectra (Fig. 1c), we have to account for the random orientation in the polycrystalline samples (with respect to the experimental beam) and average the transferred momentum in the simulations over the Brillouin zone at constant magnitude $q$, thus we only report averages over all momentum directions $e_\mathbf{q}$. Figure 3c shows that the resulting spectra are dominated by a main peak $F_1$ that is complemented by a much smaller shoulder on the high-energy side ($F_2$). With increasing $q$ values, the energy $E_1$ shifts to the red. It confirms the origin and strength of the experimental dispersion and is analyzed in more detail for the simulations. We find that most of this energy dispersion is due to the π stacking of molecules via an aggregation effect (blue shift at $q = 0$). We can use a somewhat simplifying picture to illustrate the basic mechanism at work. This picture consists of the transition dipole–transition dipole coupling model between the chromophores (Fig. 3d–f), which allows us capturing qualitatively the essential physics. We emphasise that the simulations contain a much more refined description based on the transition densities between the HOMO and LUMO orbitals of both molecules. As a result of the molecular arrangement within the stacks, triggered by the donor–acceptor character of QM1, the transition dipole coupling is positive. The increase in the wave number $q$ turns the positive dipolar coupling (at $q = 0$) into an anti-coupling dipole at the largest accessible $q$ values whose energy is then red-shifted as compared to the uncoupled case (e.g. at the X point in Fig. 3a). The sum of both adds to a total energy shift of 1 eV, which is very close to the experimental dispersion of 1 eV in Fig. 1. Despite the redshift at larger $q$, a derived feature ($F_1'$ in Fig. 3c) remains and is associated to the bands in neighboring Brillouin zones as a result of the spherical average over a larger sphere in $k$ space. Finally, we note that the origin of $F_2$ is due to CT–FE coupling (by transfer integrals), which makes the CT states between 2.7 and 3 eV, which are usually dark, accessible to EELS.

A different picture is obtained at higher excitation energies where the unconventional experimental features $F_3$ and $F_4$ were found in EELS (Fig. 1). They can be understood by including quadrupole and octupole transitions in the model. Starting from the single QM1 molecules, we first search for sizeable transition quadrupole (TQ) moments in the set of molecular pair states and find that the lowest states do not exhibit such transitions. However, the excitonic transition from the HOMO to the LUMO + 3 exhibits a sizable TQ moment. From molecular gas-phase simulations, we find their exciton energy 1.9 eV above the $E_1$ state (see Supplementary Table 2) and show these new states in the simulated spectra in Fig. 5a, which is based on the generalization of Eq. (3) to this higher multipole moment

$$\text{Im }\epsilon^M(\omega,\mathbf{q}) = \frac{4}{\Omega}\frac{e_0^2\eta}{\varepsilon_0}\left\{\frac{\left|\sum_j\frac{1}{2}e_\mathbf{q}\cdot\mathbf{Q}_j^{exi}e_\mathbf{q}|\mathbf{q}|c_{exi,\mathbf{q}}(\mathbf{r}_j,0)\right|^2}{(E_{exi}(\mathbf{q}) - \hbar\omega)^2 + \eta^2}\right\},$$
(4)

in which $\mathbf{Q}_j^{exi}$ is the TQ moment for the excitonic transition and $E_{exi}(\mathbf{q})$ its energy. Figure 5a clearly shows that the TQ feature $F_3$ emerges with increasing $q$. The peak $F_3$ includes the dominating HOMO to LUMO + 3 transition and smaller contributions from exciton states that are just below and above in energy, which yields an overall broad band between 2.6 and 4 eV, which is fully consistent with the experimental observations.

This interesting $q$-dependent intensity can be rationalized with Eq. (4). Indeed, theory predicts the absolute intensity of the TQ excitons to have an additional $|\mathbf{q}|^2$-dependent prefactor as compared to dipolar transitions. The comparison to the experimental spectra in Fig. 1 reveals that $F_3$ increases relative to $F_1$ in consistence with Eq. (4). In the opposite limit ($q \to 0$), the TQ peak disappears in the EELS

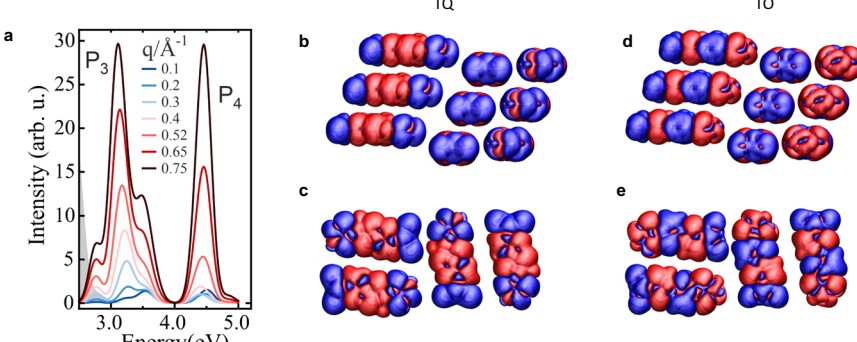

**Fig. 5 | Excitonic properties at higher excitation energies. a** Simulated EELS spectra at high excitation energy due to non-TD excitons (solid lines). $P_3$ and $P_4$ emerge due to TQ interaction and TO interaction, respectively (see main text). The position of the TD excitons at lower energy is indicated by a shaded gray area. **b** and **c** Isosurface plot of TQ density of the involved transitions proves dominating quadrupolar character with negative ends and positive center. The spacing between molecules has been increased for clarity of the figure. **d** and **e** Isosurface plot of TO density of the 11th molecular exciton.

spectra and is not visible in the UV−vis absorption spectra, all of which confirms our reasoning.

We finally investigate the origin of peak $F_4$, in Fig. 5a. It is obtained by evaluating the octupole formula

$$\operatorname{Im}\epsilon^{\mathrm{M}}(\omega,\mathbf{q}) = \frac{4}{\Omega}\frac{e_0^2\eta}{\varepsilon_0}\left\{\frac{\left|\sum_j\frac{1}{3!}\boldsymbol{e}_{\mathbf{q}}\cdot\left(\mathbf{O}_j^{\mathrm{exi}}\cdot\boldsymbol{e}_{\mathbf{q}}\right)\cdot\boldsymbol{e}_{\mathbf{q}}|\mathbf{q}|^2 c_{\mathrm{exi},\mathbf{q}}(\mathbf{r}_j,0)\right|^2}{\left(E_{\mathrm{exi}}(\mathbf{q})-\hbar\omega\right)^2+\eta^2}\right\}, \quad (5)$$

in which $\mathbf{O}_j^{\mathrm{exi}}$ is the involved transition-octupole moment of the exciton under investigation. The strongest contribution is obtained from the 11th exciton with energy at 4.44 eV (cf. Supplementary Table 2), which we call transition-octupole (TO) exciton henceforth (it has only weak TQ contributions). The band dispersions of these TQ and TO excitons are significantly smaller than for the TD exciton, which is seen in the weaker $q$ shift of the simulated EELS bands. This also explains the weak energy shift in the thin film EELS spectra at higher energies (~4.5 eV) in Fig. 1c. Fig. 5d and e finally illustrate the nature of the transition charge of the TO exciton.

## Discussion

In summary, we have studied the intriguing behavior of excitons in highly interesting organic materials in crystalline and thin-film morphologies. The extremely broad exciton bands at low energy are enabled by close contact geometries between tightly π-stacking molecular planes and strong interactions, while the indirect gap can yield excitons with a very long lifetime because they are not directly coupled to the ground state, potentially leading to unprecedentedly high concentrations upon continuous excitation. Furthermore, the remarkable anisotropy should lead to a strongly directed transport of excitons, which is interesting for controlling the main transport direction by molecular design and which is difficult to achieve in 2D herring-bone stacked polythiophenes or acenes. A survey of previously known systems suggests that only a small fraction of structures have the potential for oriented, i.e. quasi-one-dimensional transport characteristics[41]. In this context, we also note that a strict one-dimensionality of transport channels, on the other hand, could also be disadvantageous because, in the presence of disorder, it is known that quantum localization is enhanced in low dimensions[42]. This is similar for charges in organic crystals in the presence of dynamic disorder[43−46], while in the case of excitons, the often more long-range coupling could be less effective for localization[12,47], thus calling for future research of a well-tuned strategy.

If we were to estimate a characteristic time scale for the transfer between two neighboring molecules in one stack from the half-band width, we would obtain extremely short times of $\tau = 2\hbar/B \sim 1$ fs. Note

that $\tau$ would represent the time scale of the spread of excitonic quantum coherence between neighboring sites. It should not be mistaken as the motion of a localized exciton like in a hopping event[48] between two molecules because, as discussed above, the exciton should be delocalized over a large number of molecules. The combination of a long lifetime and large diffusivity further suggests a long diffusion length.

These features are a blueprint for related molecular structures, to further explore the tunability of excitonic band structures upon chemical modifications, for instance in going from the present donor−acceptor (D−A) structure motif to compounds with A−D−A motifs of non-fullerene acceptors that are studied in organic solar cells. Indeed non-fullerene acceptor materials such as Y6[49] exhibit an even larger oscillator strength. Larger oscillator strengths are also possible in other dye molecules[50] that should even further amplify these effects.

While the strong Coulomb interaction was perceived as a challenge in the research of organic semiconductors in the past, we demonstrate that its enabling role can be extended from tuning electronic properties[51−53] to tuning excitonic properties. Given that the excitonic coupling energies are comparable to gap energies and that corresponding time scales enter the attosecond regime[54], in which the vibrational dynamics are still inactive, the results offer intriguing perspectives for a controlled exciton motion in a super-diffusive regime. Indeed, the wide bands in combination with the low relaxation energy for the TD excitons of 130 meV (see e.g. refs. 55−58 for reference values) suggest an interesting transport regime. Note that the low-frequency vibrations that are responsible for the dynamic disorder can still be relevant for determining the localization/delocalization balance of the excitons, whose quantification goes beyond the scope of the present paper. This quantification and exciton transport are just two points of interest for future work. In addition, wide bands can also make excitons more resilient to disorder, which can directly increase the photocurrent in photovoltaic applications[59].

## Methods
### Synthesis
The QM1 material was prepared according to the literature[21].

### Crystal growth
The crystals were grown using a physical vapor transport (PVT) furnace with a controlled temperature gradient and a pressure of the order of $10^{-5}$ mbar. The sublimation temperature of the QM1 material was 295 °C. The growth process was conducted in the dark and took 6 days in order to get ~500 μm by ~30 μm thin stripe-like crystals.

## Crystal structure

X-ray diffraction data collection was carried out at the BESSY storage ring (BL14.2, Berlin-Adlershof, Germany)[60]. XDSAPP2.0 suite was employed for data processing[61,62]. The structure was solved by direct methods and refined by SHELXL-2018[63]. Hydrogen atoms were added geometrically, and refined with a riding model. The crystal data are presented in Supplementary Table 1.

CCDC 2172187 contains the supplementary crystallographic data for this paper. These data can be obtained free of charge from The Cambridge Crystallographic Data Centre via www.ccdc.cam.ac.uk/data_request/cif.

## Electron energy loss spectroscopy

The spectroscopic investigations using electron energy-loss spectroscopy (EELS) were carried out using a dedicated 172 keV spectrometer described in detail elsewhere[64,65].

Thin films of QM1 with a thickness of about 100 nm have been prepared by thermal evaporation of powder material and deposition onto KBr single crystals under high vacuum conditions. Free-standing films were obtained by dissolution of the KBr substrate in distilled water. The floated-off films were mounted onto standard electron microscopy grids and finally transferred into the spectrometer. Prior to the measurements of the electronic excitation spectra, electron diffraction profiles were taken, which showed that the films were essentially polycrystalline, and the resulting diffraction pattern is consistent with what one would expect from the single-crystal data[20]. In addition, the film morphology has been investigated and discussed previously (ref. 17 of the manuscript), films consist of nanowires that grow when the molecules are deposited using thermal evaporation, equivalent to the process of crystal growth. Thus, the polycrystalline films represent the crystal structure that is the basis of the calculations.

## Exciton model

Within the Frenkel exciton model, the Hamiltonian reads in an intuitive site representation, using electron and hole positions $|\mathbf{r}_{hole}, \mathbf{r}_{electron}\rangle$,

$$
\begin{aligned}
H_{ME} = & \sum_{\mathbf{n},j} E_{ME} |\mathbf{n}+\mathbf{r}_j, \mathbf{n}+\mathbf{r}_j\rangle \langle \mathbf{n}+\mathbf{r}_j, \mathbf{n}+\mathbf{r}_j| \\
& + \sum_{\mathbf{n},j,\mathbf{n}',j'} J(\mathbf{n},j,\mathbf{n}'j') |\mathbf{n}+\mathbf{r}_j, \mathbf{n}+\mathbf{r}_j\rangle \langle \mathbf{n}'+\mathbf{r}_{j'}, \mathbf{n}'+\mathbf{r}_{j'}|
\end{aligned}
\tag{6}
$$

in which $E_{ME}$ is the local excitation energy and $J(\mathbf{n},j,\mathbf{n}'j')$ describes the (statically screened) Coulomb interaction which can be expressed with transition densities $\eta_{exi}(\mathbf{r})$. Because of the small distances between the centers of the transition charges (as compared to the size of the transition charge distribution), this cannot be approximated by the conventional dipolar coupling between molecules and the $\eta_{exi}(\mathbf{r})$ is used in the present situation.

The extended exciton model includes additional CT excitons for HOMO and LUMO states. The above ME Hamiltonian is therefore extended to

$$
\begin{aligned}
H &= H_{ME} + H_{CT} + H_{coupl} \\
H_{CT} &= \sum_{\mathbf{n},j,\mathbf{s}\neq 0} E_{CT}(\mathbf{r}_j, \mathbf{s}) |\mathbf{n}+\mathbf{r}_j, \mathbf{n}+\mathbf{r}_j+\mathbf{s}\rangle \langle \mathbf{n}+\mathbf{r}_j, \mathbf{n}+\mathbf{r}_j+\mathbf{s}|
\end{aligned}
\tag{7}
$$

where $\mathbf{s}$ indicates the spatial electron-hole separation. In addition, the coupling terms between both manifolds are included as

$$
\begin{aligned}
H_{coupl} = & \sum_{\mathbf{n},j,\mathbf{s},\mathbf{a}} t_e(\mathbf{n},j,\mathbf{s},\mathbf{a}) |\mathbf{n}+\mathbf{r}_j, \mathbf{n}+\mathbf{r}_j+\mathbf{s}\rangle \langle \mathbf{n}+\mathbf{r}_j, \mathbf{n}+\mathbf{r}_j+\mathbf{s}+\mathbf{a}| \\
& + \sum_{\mathbf{n},j,\mathbf{s},\mathbf{a}} t_h(\mathbf{n},j,\mathbf{s},\mathbf{a}) |\mathbf{n}+\mathbf{r}_j, \mathbf{n}+\mathbf{r}_j+\mathbf{s}\rangle \langle \mathbf{n}+\mathbf{r}_j+\mathbf{a}, \mathbf{n}+\mathbf{r}_j+\mathbf{s}|
\end{aligned}
\tag{8}
$$

For high-energy excitons with TQ moments or TO moments we extend the manifold of states in the exciton Hamiltonian, while no CT excitons are added to this manifold.

## Data availability

The authors declare that the data supporting the findings of this study are available within the paper and its supplementary information files.

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

## Acknowledgements

We would like to thank the Deutsche Forschungsgemeinschaft (DFG) for financial support through projects OR 349/11 (F.O.), OR 349/3 (F.O.), KR 4364/4-1 (Y.K.), LE 747/60-1 (K.L.), LI 3055/3-1 (F.L.) and the Cluster of

Excellence e-conversion, grant No. EXC2089 (F.O.). Grants for computer time from the Zentrum für Informationsdienste und Hochleistungsrechnen of TU Dresden are gratefully acknowledged. F. L. appreciates the support of Dr. A.A. Popov. We would like to thank Dr. M. Weiss and his team for their support during the experiments at BESSY II at the Helmholtz-Zentrum Berlin.

## Author contributions

K.M. and F. O. performed the exciton modeling and EELS simulations. K.S.S. performed the ab initio simulations. N.G. synthesized the material. K.L. supervised the synthesis work. M.K. performed the EELS measurements. B.D. and Y.K. grew the crystals. F.L. performed X-ray studies and determined the crystal structure. F.O. wrote the main paper, with contributions from K.M., K.S.S., N.G., B.D., F.L., Y.K., K.L. and M.K.

## Funding

## Competing interests

The authors declare no competing interests.
