## [Peer Review File · Nature Communications]

Directed Exciton Transport Highways in Organic SemiconductorsFirst round reviewers' comments:

Reviewer #1 (Remarks to the Author):

The manuscript by Muller et al. concerns excitons in a molecular crystal; the most important issue is the unusually high dispersion of the excitons, which might be beneficial for high mobility in their transport properties.

From my point of view, I find the manuscript scientifically correct (except for some severe criticism, see below) and sufficiently interesting (see discussion below) to warrant publication.

Some significant criticism and questions:

1. The observation and calculation of exciton dispersion (i.e., dependence of their energy on total momentum) is, of course, not new and has been discussed for inorganic and organic semiconductor crystals in many works. The important and novel observation here is the very strong dispersion observed in this particular material, caused by the very strong transition dipole of the single molecule, which results from its linear structure with two cyano groups at one end. The critical question here, which I have difficulties to answer conclusively: is this sufficiently novel and interesting?

2. The central physical mechanism appears to be the strong hopping term of the molecular exciton from one molecule to its neighbour. This is caused by Coulomb interaction of the molecular exciton's dipole with that on the neighbouring molecule. In this work this Coulomb interaction is modeled by the simple interaction between two POINT dipoles (see Eq. (S3) in the supplement). I am not sure if this is justified, because strictly speaking Eq. (S3) is only valid at large distance between two molecules (large compared to the internal size of the molecule). But in the current material the distance between two molecules appears to be even smaller than the "size of the dipole", which comes from two excited point charges at the two ends of the molecule.

What the authors should do, in my opinion, is a proof-of-principle calculation (TDDFT, CI, or other) for a dimer of two molecules, in which the two single-molecule excitons $|1\rangle$ and $|2\rangle$ should interact and split into $|1\rangle+|2\rangle$ and $|1\rangle-|2\rangle$, with energetic splitting given by the requested interaction, and this could then be compared with Eq. (S3). Only then can the authors convincingly state that Eq. (S3) makes sense. I consider this as crucial as it constitutes the fundamental basis for the quantitative value of the dispersion, and thus for the good agreement with the experimental data.

3. The authors state that they have grown single crystals (mainly for identification of the structure?), but the spectra of Fig. 1b and 1c seem to be for some polycrystal/powder/"thin film". It is of course reasonable to assume that the microscopic structure is the same as in the single crystal, but nonetheless some more detailed discussion on the structure of the samples in Fig. 1 is necessary to justify the comparison with the theoretical spectra, which are all for a single crystal. Otherwise it would be a comparison of "apples and oranges".

4. A minor point: the discussion of hole-hole and electron-electron interaction (Eq. (S11) in the supplement) suggests that

electrons and holes are fundamentally different. Is this really intended? It seems to me that the difference is mainly in the geometrical framework, with one of the charges being located at the origin while the other is shifted in space - but it could equally well be chosen the other way around. This should be explained in a better way.

Reviewer #2 (Remarks to the Author):

I have read with much interest this paper that puts back on the front scene a technique that has been very widely applied to organic semiconductors in the late 90's, that is EELS. So the first question that came my mind when reading this manuscript is how does the picture proposed here compare to some of the earlier works performed in the Knupfer's group, namely but not only on oligophenylenes, see e.g. PRB57, R4202 or PRB61, 16538? Here the authors conclude that QM1 stands out (though it seems to me exciton bandwidths in other molecular crystals might come rather close). A more detailed comparison would, anyway, be useful to the readers and, if QM1 is indeed special, some discussion as why that is the case is called for.

If I understood it correctly, I have serious problems with the exciton model used. The authors 'coarse grained' the molecules to center points and write the interactions in a multipolar expansion. Unfortunately, there is no way this expansion possibly converges when the center-to-center separation between the molecules is smaller than the molecular size, as is the case here. In that case, what is routinely done in the field is to instead calculate the interactions between molecular transition dipoles using a multicentric expansion (3D transition densities or atom-centered transition charges). This naturally accounts for (but goes beyond) the TQ-TQ interactions in the current model. Notably the use of a point dipole approximation is known to largely overshoot the excitonic interactions, which is likely the reason why the authors have to use an artificially large dielectric constant in order to match the experimental data...

Regarding the role of CT excitons, I wonder if the authors have compared group velocities with versus without CT. It seems indeed the CT excitons are, as expected, significantly higher in energy compared to P1, so not much dispersion at the band bottom...

Though the authors are very careful in formulating their conclusions ('The combination of... suggests a long diffusion length'), there is indeed no direct evidence that QM1 'stands out' in terms of singlet exciton diffusion lengths.

Overall, though the EELS data are interesting, I am not convinced this manuscript fulfils the conditions in terms of novelty, originality and robustness of the results to warrant publication in Nature Communications.

First round point-by-point responses to reviewers' comments

Reviewer #1 (Remarks to the Author):

Reviewer comment: The manuscript by Muller et al. concerns excitons in a molecular crystal; the most important issue is the unusually high dispersion of the excitons, which might be beneficial for high mobility in their transport properties. From my point of view, I find the manuscript scientifically correct (except for some severe criticism, see below) and sufficiently interesting (see discussion below) to warrant publication.

Author reply: We thank the reviewer for this positive assessment regarding the possible publication and address the points of criticism below. We think that with the substantial changes to the manuscript we can address all points convincingly.

Reviewer comment: 1. The observation and calculation of exciton dispersion (i.e., dependence of their energy on total momentum) is, of course, not new and has been discussed for inorganic and organic semiconductor crystals in many works. The important and novel observation here is the very strong dispersion observed in this particular material, caused by the very strong transition dipole of the single molecule, which results from its linear structure with two cyano groups at one end. The critical question here, which I have difficulties to answer conclusively: is this sufficiently novel and interesting?

Author reply: The reviewer addresses the question of sufficient novelty. We would like to emphasize that the extremely strong exciton energy dispersion is the largest ever demonstrated for organic crystals, which makes the system interesting, in particular because this dispersion occurs in the optical spectral range (near infrared-visible to be precise). To make this point clearer to the reader, we make a comprehensive comparison to prior literature on organic crystals in the modified manuscript and conclude that we indeed investigate here a material with outstanding excitonic properties. This includes absolute bandwidth, low optical gap and 1D characteristics. We believe it is of high interest in current and future research on related materials.

Changes to the manuscript: We have included a new Figure 4 in the manuscript along with accompanying discussions.

Reviewer comment: 2. The central physical mechanism appears to be the strong hopping term of the molecular exciton from one molecule to its neighbour. This is caused by Coulomb interaction of the molecular exciton's dipole with that on the neighbouring molecule. In this work this Coulomb interaction is modeled by the simple interaction between two POINT dipoles (see Eq. (S3) in the supplement). I am not sure if this is justified, because strictly speaking Eq. (S3) is only valid at large distance between two molecules (large compared to the internal size of the molecule). But in the current material the distance between two molecules appears to be even smaller than the "size of the dipole", which comes from two excited point charges at the two ends of the molecule. What the authors should do, in my opinion, is a proof-of-principle calculation (TDDFT, CI, or other) for a dimer of two molecules, in which the two single-molecule excitons $|1\rangle$ and $|2\rangle$ should interact and split into $|1+\rangle|2\rangle$ and $|1-\rangle|2\rangle$, with energetic splitting given by the requested interaction, and this could then be compared with Eq. (S3). Only then can the authors convincingly state that Eq. (S3) makes sense. I consider this as crucial as it constitutes the fundamental basis for the quantitative value of the dispersion, and thus for the good agreement with the experimental data.

Author reply: The reviewer addresses a point in the modelling part of the manuscript. These simulation results are compared with the experimentally determined exciton dispersion from EELS which are not questioned.

The reviewer is right that the original theoretical model was one of the simplest models to explain the observed behavior but could be criticized because of this. The reviewer is also right that a refined model replacing the transition-dipole approach by transition densities would be more accurate. We are happy that we have the chance to implement the two suggestions of the reviewer and that the final results based on transition densities strongly corroborate the strong experimental value of the dispersion quantitatively. While we see some qualitative differences for the simulated bands, our main conclusions remain unchanged.

Overall the **EELS spectra are much better described in the improved simulations**. While this might not be to the surprise of the reviewer, we are particularly happy about this agreement. In particular, feature F_4 is now understood as transition-octupole-mediated excitation feature from high-energy excitons. We are not aware of a comparable investigation of the exciton anatomy in organic thin films or crystals and thank both reviewers for their helpful comments.

Changes to the manuscript: We have modified the manuscript by replacing the point dipole model in Eq. (S3) by the model in which all interactions are based on atom-projected transition densities. Many parts have been rewritten.

Reviewer comment: 3. The authors state that they have grown single crystals (mainly for identification of the structure?), but the spectra of Fig. 1b and 1c seem to be for some polycrystal/powder/"thin film". It is of course reasonable to assume that the microscopic structure is the same as in the single crystal, but nonetheless some more detailed discussion on the structure of the samples in Fig. 1 is necessary to justify the comparison with the theoretical spectra, which are all for a single crystal. Otherwise it would be a comparison of "apples and oranges".

Author reply: The EELS spectra as shown in Fig. 1 indeed have been measured using polycrystalline films as written in the manuscript. These films had to be taken since the available single crystals were unfortunately too small for our EELS setup. Prior to the EELS measurements, we have characterized the films with electron diffraction. The resulting diffraction pattern is consistent to what one would expect from the single-crystal data. In addition, the film morphology has been investigated and discussed previously (Ref. 17 of the manuscript), they consist of nanowires which grow when the molecules are deposited using thermal evaporation, equivalent to the process in crystal growth. Thus, the polycrystalline films represent the crystal structure that is the basis of the calculations.

In addition, we did exciton model simulations in which we included also disorder to the crystalline structure by simulating larger supercells and testing different disorder models. In short, we did study stacking faults for example and randomized transition dipoles as two examples of disorder. The former is a specific defect, the latter is more a model study. The effect of these perturbations to the simulated EELS spectra of the crystal were rather small for reasonable densities. We found that the reason in the strong dipol-dominated coupling that remains for adjacent pairs of molecules.

Changes to the manuscript: We have extended the EELS part in the "Methods" section to give these details on structural investigations.

Reviewer comment: 4. A minor point: the discussion of hole-hole and electron-electron interaction (Eq. (S11) in the supplement) suggests that electrons and holes are fundamentally different. Is this really intended? It seems to me that the difference is mainly in the geometrical framework, with one of the charges being located at the origin while the other is shifted in space - but it could equally well be chosen the other way around. This should be explained in a better way.

Author reply: The reviewer addressed a point that can be made clearer in the Supporting Information. We agree with them that there is no fundamental difference between electron and holes, but we have to select a representation for concreteness during the implementation. When introducing a distance vector between electron and hole and introducing the exciton basis, we have to choose whether the electron or the hole is in the same unit cell as the exciton reference point. That is why there is an apparent asymmetry in the formulas. Upon closer investigating this, it becomes clear that there is no physical asymmetry. As suggested by the reviewer, this representation could well be the other way around without impacting any result.

Changes to the manuscript: We have added some comments addressing this point in the Supporting Information, such as “We emphasize that the apparent asymmetry between electrons and holes is not a true physical asymmetry but is a consequence of the particular choice we have to make by introducing explicitly the distance vector between electron and hole in the pair-state notation. All results are independent of this choice.” And “Here, for concreteness of the representation, we use the hole position as the reference position for the exciton. Note that this introduces an *insignificant* asymmetry in the representation of the Hamiltonian between electrons and holes.”

Reviewer #2 (Remarks to the Author):

Reviewer comment: I have read with much interest this paper that puts back on the front scene a technique that has been very widely applied to organic semiconductors in the late 90's, that is EELS. So the first question that came my mind when reading this manuscript is how does the picture proposed here compare to some of the earlier works performed in the Knupfer's group, namely but not only on oligophenylenes, see e.g. PRB57, R4202 or PRB61, 16538? Here the authors conclude that QM1 stands out (though it seems to me exciton bandwidths in other molecular crystals might come rather close). A more detailed comparison would, anyway, be useful to the readers and, if QM1 is indeed special, some discussion as why that is the case is called for.

Author reply: We thank the reviewer for expressing their great interest in our study. Reviewer #2 asks for a comparison of the results and conclusion in our manuscript to earlier work of one of the co-authors, in which also EELS studies of organic semiconductors have been presented. Our answer is two-fold.

In particular, Reviewer #2 refers to two earlier publications: PRB57, R4202 and PRB61, 16538. In the first of these two publication, EELS has been used to study a polymer film. In this case the observed momentum dependence arises from the intra-polymer band structure of the polymer, which could be possibly related to a one-dimensional solid. This is quite different from the present manuscript where the observed exciton dispersion is caused by inter-molecular interactions (which do not play a role in the polymer case!). To our opinion, a comparison of these results in the manuscript would rather distract than help. In the second previous publication as mentioned by Reviewer #2, no EELS data are presented, so a comparison does not seem meaningful to us.

On the other hand, there is previous work where we have measured the exciton dispersion in molecular crystals caused by inter-molecular interactions, which give rise to the exciton dispersion in these materials. Examples are: Phys. Rev. B66, 035208 (2002), Phys. Rev. Lett. 98, 037402 (2007), Phys. Rev. B83, 165436 (2011), J. Chem. Phys. 136, 204708 (2012), EPL 112, 37004 (2015), AIP Advances 11, 095313 (2021), J. Phys. Chem. C 125, 12398 (2021), ACS Omega 7, 21183 (2022). We emphasize that in all previously measured cases the exciton dispersion was much smaller than in QM1 as discussed in the present manuscript. A more detailed comparison is included by including Figure 4. This comparison clarifies the outstanding behavior of QM1 where the exciton physics is governed by very strong inter-molecular coupling. We thank the reviewer for allowing us to make this comparison in the novel version of the manuscript.

Changes to the manuscript: We have extended the manuscript in order to address the outstanding role of QM1. This includes a new figure (Figure 4) where previous measurements are summarized and compared for a better overview. Data is added to the Supporting Material.

Reviewer comment: If I understood it correctly, I have serious problems with the exciton model used. The authors 'coarse grained' the molecules to center points and write the interactions in a multipolar expansion. Unfortunately, there is no way this expansion possibly converges when the center-to-center separation between the molecules is smaller than the molecular size, as is the case here. In that case, what is routinely done in the field is to instead calculate the interactions between molecular transition dipoles using a multicentric expansion (3D transition densities or atom-centered transition charges). This naturally accounts for (but goes beyond) the TQ-TQ interactions in the current model. Notably the use of a point dipole approximation is known to largely overshoot the excitonic interactions, which is likely the reason why the authors have to use an artificially large dielectric constant in order to match the experimental data...

Author reply: We thank the reviewer for the comment and helpful suggestion to use a more elaborate (and qualitatively better) model than we did in our initial submission. In the new manuscript we use the proposed transition densities for calculations of the interactions and a larger number of excitons. In addition, the large dielectric constant was lowered as suggested by the reviewer.

As a result, the strong exciton band dispersion **and large exciton bandwidth is confirmed** for the strongly dispersing low-energy excitons. This shows that our conclusions are robust. The purpose of the previous molecular multipole expansion was to maximize transparency of the model, but we fully agree on the better justification and higher accuracy of the proposed method.

There are also differences with the new model at high energy. Overall the EELS spectra are much better described in the improved simulations. We are particularly happy about this agreement. In particular, feature F_4 is now understood as transition-octupole-mediated excitation feature from high-energy excitons.

Reviewer comment: Regarding the role of CT excitons, I wonder if the authors have compared group velocities with versus without CT. It seems indeed the CT excitons are, as expected, significantly higher in energy compared to P1, so not much dispersion at the band bottom...

Though the authors are very careful in formulating their conclusions ('The combination of... suggests a long diffusion length'), there is indeed no direct evidence that QM1 'stands out' in terms of singlet exciton diffusion lengths.

Overall, though the EELS data are interesting, I am not convinced this manuscript fulfils the conditions in terms of novelty, originality and robustness of the results to warrant publication in Nature Communications.

Author reply and changes to the manuscript: Regarding the first point, we see at some points in the exciton band structure (Figure 3) evidence of the coupling, which we highlight there. In the region of strongest band velocity, ME-CT coupling is minor. Experimental evidence for ME-CT coupling manifests in feature F_2 , which we include in the discussion in the novel manuscript. Moreover, as another novelty we could identify clear octupole-transitions in feature F_4 .

We have substantially revised the modelling part according to the suggestions of Reviewer #2 and found the extremely large bandwidth to be robust. In a new figure (Figure 4) we now compare the results with prior measurements of other systems. From this comparison the outstanding behavior of QM1 is much clearer. This should visualize the novelty much better.

Second round reviewers' comments

Reviewer #1 (Remarks to the Author):

The manuscript by Muller et al. concerns excitons in a molecular crystal; the most important issue is the unusually high dispersion of the excitons, which might be beneficial for high mobility in their transport properties.

I had reviewed the manuscript in its previous version (June 2022). In my impression the paper has improved substantially, in particular with regards to my (and the other referee's) previous concerns on the validity of the microscopic origin and quantitative evaluation of the molecule-molecule-interaction.

This and most other criticisms seem to have been addressed and settled.

From my point of view, I find the manuscript scientifically correct and sufficiently interesting to warrant publication.

Reviewer #3 (Remarks to the Author):

The paper from Müller et al. represents an interesting study regarding a new system absorbing in the NIR showing a very high exciton band dispersion. They compared the findings with EELS in a successful manner, which is nice to see. I believe the work is technically well done, and in my opinion the authors addressed satisfactorily the concerns of the other reviewers. Though I would tend to join the other reviewers in expressing some concerns about its novelty (since they are targeting a high-impact journal). I do also have some other concerns about different aspects in addition to those already answered by the authors. In addition, I found the bibliography a little bit restricted. I believe the authors could put their work in better context with respect to existing literature. It would be nice if the authors could address my comments in a further round of corrections.

1) My first observation regards the section of the abstract where they talk about design principles. In the end is not quite clear to me which new design principles they achieved in this study and which ones are the authors referring to in the abstract. For instance, in the discussion section on page 12, they mention "tight packing and strong exciton interaction" or "long lifetimes", but these are all very well know design principles. I think the authors should be a bit more precise with what they mean with their new design principles.

2) My second point regards the comment about the strong anisotropy of the interactions that the authors deemed as useful because it should lead to directional exciton transport. I think the author should be careful with the fact that, although the transport might be very efficient in a single direction, strong anisotropy is known to be disadvantageous for the transport (as compared to 2D and 3D). I think the authors should acknowledge this fact in the text and extend their literature report, e.g. by citing Balzer, D. & Kassal, I. J. Phys. Chem. Lett. 14, 2155–2162 (2023), Giannini, S. et al. Nat. Commun. 13, 2755 (2022) ; and for instance the same issue was proposed for charge transport in Fratini, S., Ciuchi, S., Mayou, D., de Laisardière, G. T. & Troisi, A. Nat. Mater. 16, 998–1002 (2017) , Giannini, S., Ziogos, O. G., Carof, A., Ellis, M. & Blumberger, J Adv. Theory Simulations 3, 2070021 (2020) and Balzer, D., Smolders, T. J. A. M., Blyth, D., Hood, S. N. & Kassal, I. Chem. Sci. 12, 2276–2285 (2021).

3) One thing that I would find beneficial for reproducibility of the results and actually better understanding of the strong interactions in real space, would be a table (even in the SI) where the magnitude of ME-ME and ME-CT interactions (indicated in Fig. 2) are reported. This would make the story easier to follow. Also, with such a table the phase of the interactions would be clearly presented (this being an important issue for the actual nature of the exciton dispersion, see e.g., Hestand, N. J. et al. *J. Phys. Chem. C* 119, 22137–22147 (2015)).

4) Although the authors on page 13 brought about the argument that excitonic coupling are close to the gap energy and vibrational dynamics is still inactive, I am not sure I totally agree with it. Disorder has been shown to be important even for molecules similar to the one investigated here (see Aragó, J. & Troisi, A. *Adv. Funct. Mater.* 26, 2316–2325 (2016) and Aragó, J. & Troisi, A. *Phys. Rev. Lett.* 114, 1–5 (2015)). I think the authors should cite these works and expand the discussion about the role of disorder (for instance dynamic disorder). Also, since it is simple enough, I think the author could also calculate the exciton reorganization energy for the single molecule and give a number for it. This, along the excitonic coupling in the new table I have mentioned before, would already give an idea about what to expect in terms of exciton diffusivity. See for instance: Nematiram, T., Padula, D. & Troisi, A. *Bright Frenkel Excitons in Molecular Crystals: A Survey. Chem. Mater.* 33, 3368–3378 (2021) and Xie, X. & Troisi, A. *Identification via Virtual Screening of Emissive Molecules with a Small Exciton-Vibration Coupling for High Color Purity and Potential Large Exciton Delocalization. J. Phys. Chem. Lett.* 4119–4126 (2023).

Reviewer #4 (Remarks to the Author):

The authors report electron energy loss spectra (EELS) for quinoid merocyanine thin films and crystals showing large excitonic bandwidths and, simultaneously, low band gap in the VIS-near infrared. The combination of these two properties is unusual (as evidenced by Table S3) and potentially beneficial for optoelectronic applications. The excitonic band structure and the EELS spectra are simulated and the latter well reproduced using (TD)DFT parametrized excitonic Hamiltonians.

The work is well presented, some of the conclusions made are not well justified and should be revised. I recommend publication after the points below have been addressed. Further review is not necessary.

1. page 3: "making it difficult to observe delocalized excitons at room temperature and to control their pathways." Brief discussion and citation of recent works in the literature where exciton delocalization was predicted or observed in acenes, non-fullerene acceptors or polythiophene

nanofibres would be appropriate:

Sneyd et al *JPCL* 13, 6820 (2022)

Giannini et al *Nat Commun* 13, 2755 (2022)

Alvertis et al *PRL* 130, 086401 (2023)

Alvertis et al *PRB* 081122(R) (2020)

2. The full excitonic coupling between molecules is comprised of the Coulomb term (that includes the transition densities), exchange and overlap terms. The authors only consider the Coulomb term

(Eq. S3) The authors should discuss the importance of the exchange and overlap terms given that the molecules are very closely spaced.

There are a few statements in the Discussion section that should be revised.

3. page 12: "We note that when estimating the time scale for this motion between two neighboring molecules in one stack from the half band width, we obtain extremely short times of $\tau = 2\hbar/B \sim 1$ fs."

I don't think this estimate is meaningful. If excitonic couplings are as large as reported the exciton will no longer be localized on a single molecule - in this case any notion of "motion between two neighbouring molecules" is misleading and should be avoided.

4. page 12: "The combination of long lifetime and large diffusivity further suggests a long diffusion length."

The large bandwidth does not immediately imply large diffusivity. The latter depends on a number other important properties not reported in the paper, e.g., the magnitude of reorganization energy in excited state and off-diagonal exciton-phonon coupling. If the latter two are large, diffusivity may still be low despite the large bandwidth/excitonic couplings. It would be insightful if the authors could calculate the reorganization energy in the S1 excited state and compare to excitonic coupling.

5. The authors should report the excitonic couplings and transfer integrals for holes and electrons used to set up the excitonic Hamiltonian.

Second round point-by-point responses to reviewer comments

Reviewer #1 (Remarks to the Author):

Reviewer comment: *The manuscript by Muller et al. concerns excitons in a molecular crystal; the most important issue is the unusually high dispersion of the excitons, which might be beneficial for high mobility in their transport properties.*

I had reviewed the manuscript in its previous version (June 2022). In my impression the paper has improved substantially, in particular with regards to my (and the other referee's) previous concerns on the validity of the microscopic origin and quantitative evaluation of the molecule-molecule-interaction. This and most other criticisms seem to have been addressed and settled.

From my point of view, I find the manuscript scientifically correct and sufficiently interesting to warrant publication.

Authors' reply: We thank the reviewer for noticing the substantial improvement of the manuscript, expressing their interest in the study and for the recommendation of publication.

Reviewer #3 (Remarks to the Author):

Reviewer comment: *The paper from Müller et al. represents an interesting study regarding a new system absorbing in the NIR showing a very high exciton band dispersion. They compared the findings with EELS in a successful manner, which is nice to see. I believe the work is technically well done, and in my opinion the authors addressed satisfactorily the concerns of the other reviewers. Though I would tend to join the other reviewers in expressing some concerns about its novelty (since they are targeting a high-impact journal). I do also have some other concerns about different aspects in addition to those already answered by the authors. In addition, I found the bibliography a little bit restricted. I believe the authors could put their work in better context with respect to existing literature. It would be nice if the authors could address my comments in a further round of corrections.*

Authors' reply: We thank the reviewer for the positive assessment that the work is “well done”. We are happy to extend the list of references to underline the novelty of our work and put it into a broader context with respect to existing theoretical work. We understand that the reviewer's comments aim at improving the manuscript in a round of corrections towards publication and the remaining points are addressed below. The original 50 references have been extended to 65.

Reviewer comment: *1) My first observation regards the section of the abstract where they talk about design principles. In the end is not quite clear to me which new design principles they achieved in this study and which ones are the authors referring to in the abstract. For instance, in the discussion section on page 12, they mention “tight packing and strong exciton interaction” or “long lifetimes”, but these are all very well know design principles. I think the authors should be a bit more precise with what they mean with their new design principles.*

Authors' reply: We have removed the word “design principle” to avoid such misunderstandings. Donor-acceptor type molecules such as QM1 are different to the reference systems with comparable excitonic bandwidths (mainly the polythiophenes in Fig. 4 of the manuscript show comparable values). In the manuscript we emphasize the differences in the molecular properties (dipolar, D-A type), crystal structure, and excitonic and optical properties, including different anisotropy and optical gap. This results directly from the molecular structure and can be further explored in the future.

Changes to the manuscript: We have changed the sentence in the abstract to: “The observations and detailed understanding of the results yield blueprints for the design of molecular structures in which similar molecular features might be used to further explore the tunability of excitonic bands...” A related sentence in the discussion section was changed accordingly.

Reviewer comment: 2) *My second point regards the comment about the strong anisotropy of the interactions that the authors deemed as useful because it should lead to directional exciton transport. I think the author should be careful with the fact that, although the transport might be very efficient in a single direction, strong anisotropy is known to be disadvantageous for the transport (as compared to 2D and 3D). I think the authors should acknowledge this fact in the text and extend their literature report, e.g. by citing Balzer, D. & Kassal, I. J. Phys. Chem. Lett. 14, 2155–2162 (2023), Giannini, S. et al. Nat. Commun. 13, 2755 (2022) ; and for instance the same issue was proposed for charge transport in Fratini, S., Ciuchi, S., Mayou, D., de Laissardière, G. T. & Troisi, A. Nat. Mater. 16, 998–1002 (2017) , Giannini, S., Ziogos, O. G., Carof, A., Ellis, M. & Blumberger, J Adv. Theory Simulations 3, 2070021 (2020) and Balzer, D., Smolders, T. J. A. M., Blyth, D., Hood, S. N. & Kassal, I. Chem. Sci. 12, 2276–2285 (2021).*

Authors’ reply/Changes to the manuscript: We are fully aware of this aspect that electronic quantum localization is stronger in 1D than in higher dimensions, which has been studied beginning with the work of P.W. Anderson, and is widely known as Anderson localization – a fact that has since been observed in different fields of physics. We now mentioned this in our article by pointing to the more recent references mentioned by the reviewer.

Having a single transport direction in the above strict sense is not the point that we want to make. The aspects we emphasize here are the following ones: It is useful when the direction of the transition dipole moments can be modified, such as in OPV devices. It is also useful when excitons move in the right direction to reduce losses. This cannot be achieved by an isotropic characteristic. Therefore, being able to tune the directionality for exciton motion by choice of the material would be useful. In this sense the present system is interesting because it is different from the herringbone materials studied before. A more balanced discussion of these points is now contained in the Discussion section of the manuscript.

Reviewer comment: 3) *One thing that I would find beneficial for reproducibility of the results and actually better understanding of the strong interactions in real space, would be a table (even in the SI) where the magnitude of ME-ME and ME-CT interactions (indicated in Fig. 2) are reported. This would make the story easier to follow. Also, with such a table the phase of the interactions would be clearly presented (this being an important issue for the actual nature of the exciton dispersion, see e.g., Hestand, N. J. et al. J. Phys. Chem. C 119, 22137–22147 (2015)).*

Authors’ reply/Changes to the manuscript: We have put four tables in the SI (Table S3-S6) containing such information to complement the updated Figure S2 in which the largest transfer integrals are indicated for reproducibility.

Reviewer comment: 4) *Although the authors on page 13 brought about the argument that excitonic coupling are close to the gap energy and vibrational dynamics is still inactive, I am not sure I totally agree with it. Disorder has been shown to be important even for molecules similar to the one investigated here (see Aragó, J. & Troisi, A. Adv. Funct. Mater. 26, 2316–2325 (2016) and Aragó, J. & Troisi, A. Phys. Rev. Lett. 114, 1–5 (2015)). I think the authors should cite these works and expand the discussion about the role of disorder (for instance dynamic disorder). Also, since it is simple enough, I*

think the author could also calculate the exciton reorganization energy for the single molecule and give a number for it. This, along the excitonic coupling in the new table I have mentioned before, would already give an idea about what to expect in terms of exciton diffusivity. See for instance: Nemataram, T., Padula, D. & Troisi, A. Bright Frenkel Excitons in Molecular Crystals: A Survey. Chem. Mater. 33, 3368–3378 (2021) and Xie, X. & Troisi, A. Identification via Virtual Screening of Emissive Molecules with a Small Exciton-Vibration Coupling for High Color Purity and Potential Large Exciton Delocalization. J. Phys. Chem. Lett. 4119–4126 (2023).

Authors' reply: We thank the reviewer for his/her personal interest in a more detailed discussion of this point. We have calculated and included the value of the reorganization energy for a single molecule (the relaxation energy λ) which is 130 meV and 128 meV, depending on the charging state. This is at the lower end when compared to excitons in other small molecules. [see e.g. Aragó, J. & Troisi, A. Adv. Funct. Mater. 26, 2316–2325 (2016) and Phys. Rev. Lett. 114, 026402 (2015), Vandewal, K. JACS 139, 1699 (2017), Xie, X. & Troisi, A. J. Phys. Chem. Lett. 4119–4126 (2023)] When we take the bandwidth as a reference energy, λ is much smaller, suggesting the electronic time scale to be much faster. Our comment that “vibrational dynamics is still inactive” is therefore substantiated by these numbers.

The low-frequency oscillations that are responsible for dynamic disorder have even lower oscillation time scale. The reviewer mentions that the modes are still relevant for determining the localization/delocalization balance and diffusivity, which to some extent is indeed expected but whose quantification goes beyond the scope of the present paper. It is a point of interest for future work. The same holds for the exciton diffusivity for which we prefer not to speculate on values.

Changes to the manuscript: We have included the value of the relaxation energy for a single molecule and the references in the main text as requested. We also mention the role of dynamic disorder in the manuscript. The last two papers were included in the appropriate context.

Reviewer #4 (Remarks to the Author):

Reviewer comment: *The authors report electron energy loss spectra (EELS) for quinoid merocyanine thin films and crystals showing large excitonic bandwidths and, simultaneously, low band gap in the VIS-near infrared. The combination of these two properties is unusual (as evidenced by Table S3) and potentially beneficial for optoelectronic applications. The excitonic band structure and the EELS spectra are simulated and the latter well reproduced using (TD)DFT parametrized excitonic Hamiltonians. The work is well presented, some of the conclusions made are not well justified and should be revised. I recommend publication after the points below have been addressed. Further review is not necessary.*

Authors' reply: We thank the reviewer for the positive assessment of our work and recommendation of publication. We are happy to improve the justification of the conclusions/and or improve the conclusions themselves as detailed below.

Reviewer comment: *1. page 3: “making it difficult to observe delocalized excitons at room temperature and to control their pathways.” Brief discussion and citation of recent works in the literature where exciton delocalization was predicted or observed in acenes, non-fullerene acceptors*

or polythiophene, nanofibres would be appropriate: Sneyd et al JPCL 13, 6820 (2022), Giannini et al Nat Commun 13, 2755 (2022), Alvertis et al PRL 130, 086401 (2023), Alvertis et al PRB 081122(R) (2020)

Authors' reply/ Changes to the manuscript: We have updated the list of references for a better coverage of recent literature and extended the text in this part of the manuscript, as suggested, and the discussion of delocalization.

Reviewer comment: 2. *The full excitonic coupling between molecules is comprised of the Coulomb term (that includes the transition densities), exchange and overlap terms. The authors only consider the Coulomb term (Eq. S3) The authors should discuss the importance of the exchange and overlap terms given that the molecules are very closely spaced.*

Authors' reply/ Changes to the manuscript: The full excitonic Hamiltonian, when expressed in a basis, consists of matrix elements, some of which are due to the Coulomb interaction and some are single-particle terms. The Coulomb-mediated contributions can be further characterized as mentioned by the reviewer (direct Coulomb, exchange etc.). We include a brief discussion of the smaller terms in the SI. The used terms in our modelling stem from *intramolecular* transition densities, which are known to be much larger (at least for the strong transitions that are of interest here) than the mentioned terms due to non-local (*intermolecular*) density matrices. It is known that these decay exponentially. The same holds true for their overlaps.

Reviewer comment: *There are a few statements in the Discussion section that should be revised. 3. page 12: "We note that when estimating the time scale for this motion between two neighboring molecules in one stack from the half band width, we obtain extremely short times of $\tau = 2\hbar/B \sim 1$ fs." I don't think this estimate is meaningful. If excitonic couplings are as large as reported the exciton will no longer be localized on a single molecule - in this case any notion of "motion between two neighbouring molecules" is misleading and should be avoided.*

Authors' reply: We thank the reviewer for this hint. By giving this time scale we don't imply a motion of localized excitons between to sites (like in a hopping-type event, see *J. Phys. Mater. 5, 024001 (2022)*). This is a misunderstanding. Still the given time estimate is useful and should be understood as a measure of the spread of coherence. We agree that delocalization is a suitable concept for such values of the bandwidth.

Changes to the manuscript: We have revised this statement accordingly: *"If we were to estimate a characteristic time scale for the transfer between two neighboring molecules in one stack from the half band width, we would obtain extremely short times of $\tau = 2\hbar/B \sim 1$ fs. Note that τ would represent the time scale of the spread of excitonic quantum coherence between neighboring sites. It should not be mistaken as the motion of a localized exciton like in a hopping event⁴⁸ between two molecules because, as discussed above, the exciton should be delocalized over a large number of molecules."*

Reviewer comment: 4. page 12: *"The combination of long lifetime and large diffusivity further suggests a long diffusion length." The large bandwidth does not immediately imply large diffusivity. The latter depends on a number other important properties not reported in the paper, e.g., the magnitude of reorganization energy in excited state and off-diagonal exciton-phonon coupling. If the latter two are large, diffusivity may still be low despite the large bandwidth/excitonic*

couplings. It would be insightful if the authors could calculate the reorganization energy in the S1 excited state and compare to excitonic coupling.

Authors' reply Indeed, the large bandwidth does not imply in any case a high diffusivity but here the bandwidth is very large and hence a strong driving force for large diffusivity. Only if this is counteracted by a competing effect, the diffusivity could remain low, as mentioned by the reviewer. To check this, we have calculated and added the relaxation energy (single molecule reorganization energy) of the lowest exciton state and extended the discussion on this point. The relaxation energy of 130 meV is much lower than the bandwidth of 1.3 eV, hence the statement "The combination of long lifetime and large diffusivity further suggests a long diffusion length." is correct. Off-diagonal exciton-phonon coupling is less effective because of the Coulomb nature of the coupling, which is different to charges, where the overlap of wave function with their nodal structure is relevant.

Changes to the manuscript: The relaxation energy is calculated and added to the manuscript text.

Reviewer comment: 5. *The authors should report the excitonic couplings and transfer integrals for holes and electrons used to set up the excitonic Hamiltonian.*

Authors' reply/ Changes to the manuscript: We have included a list of the transfer integrals and excitonic couplings in the SI (tables S3-S6).

Third round reviewers' comments

Reviewer #3 (Remarks to the Author):

The authors answered all my comments in a clear manner. I suggest publication of the paper is its current form.